# Unraveling the Genetic Heartbeat: Decoding Cardiac Involvement in Duchenne Muscular Dystrophy

**DOI:** 10.3390/biomedicines13010102

**Published:** 2025-01-04

**Authors:** Valeria Novelli, Francesco Canonico, Renzo Laborante, Martina Manzoni, Alessandra Arcudi, Giulio Pompilio, Eugenio Mercuri, Giuseppe Patti, Domenico D’Amario

**Affiliations:** 1Centro Cardiologico Monzino IRCCS, 20138 Milan, Italy; 2Thoracic-Cardiovascular Department, Azienda Ospedaliero-Universitaria Maggiore della Carità, 28100 Novara, Italyalessandra.arcudi@gmail.com (A.A.);; 3Fondazione Policlinico Universitario Agostino Gemelli IRCCS, Catholic University of the Sacred Heart, 00168 Rome, Italy; 4Department of Translational Medicine, Università del Piemonte Orientale, via Solaroli, 17, 28100 Novara, Italy

**Keywords:** next-generation sequencing, Duchenne muscular dystrophy, dilated cardiomyopathy, genetic variants, heart failure

## Abstract

Cardiomyopathy represents the most important life-limiting condition of Duchenne muscular dystrophy (DMD) patients after the age of 20. Genetic alterations in the DMD gene result in the absence of functional dystrophin protein, leading to skeletal/cardiac muscle impairment. The DMD incidence is one in 5000 live male births. Identifying the genetic background, in addition to DMD disease-causing variants, is one of the unmet needs in understanding the cardiac disease’s pathogenetic mechanisms and its prognostic implications. The clinical scenario is made even more intricate by the difficulty in predicting the onset and progression of cardiomyopathy, as no clear genotype/phenotype correspondence has been found thus far. The evaluation of genes involved in the onset of primary cardiomyopathies could explore the hypothesis that changes in cytoskeletal and sarcomeric protein function are the modulators of ventricular dysfunction in DMD patients. In the last decade, with the advent of next-generation sequencing (NGS) technology, many disease-causing genes and modifiers have been identified. Assessing the genetic origin of the phenotypic variability of the disease in both the onset and progression of cardiomyopathy in DMD would be extremely helpful in managing these patients. This review article aims to spotlight the genetic background associated with Cardiomyopathy in DMD patients toward a more predictive personalized model of care.

## 1. Introduction

Duchenne muscular dystrophy (DMD) is caused by mutations in the *DMD* gene (HGNC:2928), leading to the absence of functional dystrophin and impairing both skeletal and cardiac muscle function. A recent systematic review on the global epidemiology of DMD reported a birth prevalence of 19.8 per 100,000 live male births, with survival rates varying significantly worldwide due to differences in adherence to care standards in developing countries [1]. DMD and Becker muscular dystrophy (BMD) are both characterized by a loss of dystrophin expression, with total loss in DMD and partial loss in BMD. The degree of dystrophin deficiency influences the clinical course, with DMD presenting a more severe, early-onset progression, while BMD typically has a later and less severe onset [2]. DMD is a progressive neuromuscular disorder that severely impacts pulmonary and cardiac function. Children with DMD typically present with a waddling gait, lordotic posture, calf hypertrophy, and a positive Gower sign. Muscle deterioration begins between ages six and eight, with noticeable lordosis and scoliosis. As respiratory function declines and infections increase, respiratory failure often becomes a critical concern later in life [2]. DMD is disabling and life-limiting, significantly affecting both the patient and their family. The reduced physical capacity and progressive nature of the disease contribute to a compromised quality of life [3]. However, advances in respiratory management and early steroid treatment have improved life expectancy, making cardiac complications more prominent in the natural history of the disease.

Given the progressive nature of both muscular and cardiac degeneration, systematic and regular cardiac assessments are essential to monitor heart function and guide appropriate interventions. Early detection of myocardial fibrosis and other signs of cardiac dysfunction is crucial, as these often precede clinical heart failure (HF). Routine imaging including serial echocardiograms can provide valuable insight into changes in left ventricular (LV) function, wall motion, and chamber size, allowing for the early identification of abnormalities. As the disease progresses, cardiac magnetic resonance imaging (MRI) becomes increasingly important in assessing myocardial fibrosis, tissue damage, and ventricular remodeling. Both imaging techniques are essential in tracking the trajectory of cardiac involvement and tailoring interventions at different stages of the disease.

In the early stages of DMD, cardiac involvement is typically subclinical, with myocardial fibrosis beginning before the development of overt HF. Serial echocardiograms starting around ages 6–8 years can help identify subtle changes in cardiac function, allowing for the early initiation of cardioprotective therapies such as ACE inhibitors, beta-blockers, or diuretics. Early intervention has been shown to improve long-term outcomes and delay progression to more severe HF. As patients transition into adolescence, the incidence of dilated cardiomyopathy (DCM) and arrhythmias increases, making routine imaging crucial for risk assessment [2,3,4,5]. Serial echocardiograms, complemented by cardiac MRI, can monitor for structural changes like LV dilation and myocardial fibrosis, providing a more comprehensive understanding of the heart’s evolving function. These assessments enable clinicians to initiate appropriate pharmacological therapies before irreversible heart damage occurs.

For older adolescents and young adults, where cardiomyopathy is universally present, managing HF and preventing arrhythmias becomes the main focus. Despite advances in pharmacologic treatments in the adult heart failure field, there is still a critical gap in DMD care, mainly in the transition phase, due to the limited evidence with the utilization of novel therapies such as sodium-glucose cotransporter 2 inhibitors (SGLT2is) and angiotensin receptor-neprilysin inhibitors (ARNIs) in DMD setting. While these agents have shown promise in treating HF in broader populations, their efficacy and safety in DMD patients are still under investigation. In addition, advanced HF management options like mechanical circulatory support devices (i.e. left ventricular assist devices (LVADs)) have not been systematically investigated in DMD patients, although the limited experiences reported are promising. LVADs could potentially benefit patients with end-stage HF, but the intra- and peri-procedural risks that are specific of the DMD populations, among which, muscle weakness, respiratory mechanics, and lordosis must be carefully evaluated before proposing the implant. Furthermore, heart transplant is considered a relative contraindication in DMD due to the underlying genetic mutation affecting not only skeletal and cardiac muscle but also post-transplant immune function, making long-term graft survival more challenging [6].

Given the individualized progression of cardiac involvement in DMD, influenced by genetic factors and disease stage, a personalized approach to cardiac monitoring and management is essential. Regular, systematic imaging with serial echocardiograms and MRI, combined with clinical evaluations, can provide critical insights into the progression of cardiac dysfunction and guide the initiation of targeted therapies. These imaging modalities are key to identifying early signs of myocardial damage, allowing for timely interventions that can improve cardiac outcomes and overall quality of life. Furthermore, expanding research into novel treatments like SGLT2 inhibitors, ARNI, and advanced therapies like LVADs is essential to filling the gaps in care for DMD patients. By improving our understanding of these therapies and their applicability in DMD, we can provide more effective, personalized care to mitigate the progression of HF, reduce the risk of arrhythmias, and ultimately improve the lifespan and quality of life of individuals with DMD.

## 2. Genetic and Pathophysiological Background in DMD Patient

The *DMD* gene (OMIM 300377) is one of the largest known genes, spanning approximately 2.5 Mb on the X chromosome (Xp21.1). Pathogenic variants in the *DMD* gene, predominantly deletions and duplications, result in either the absence or dysfunction of dystrophin, a crucial cytoskeletal protein. This loss leads to progressive muscle degeneration, with substantial phenotypic variability that is determined by the type of genetic alteration and its impact on dystrophin expression. In cases where disease progression is milder than anticipated for DMD, differential diagnoses such as intermediate muscular dystrophy (IMD) or BMD should be considered [7]. Deletions represent the majority of mutations (over 70%), often leading to premature stop codons, while point mutations, small deletions, or insertions account for approximately 20% of cases. The remaining 5–15% of mutations are typically duplications, with the majority of deletions and duplications occurring in exons 44–55 and 3–9 of the *DMD* gene [8]. Although deletions, duplications, and small mutations in the *DMD* gene account for more than 99% of DMD/BMD cases, rarer genomic rearrangements involving translocations between the X chromosome and autosomes have also been reported [3].

Recently, attention has expanded to the role of deep intronic variants in *DMD*. For instance, two deep intronic mutations, c.8669-19_8669-24del and c.6439-1016_6439-3376del, have been characterized in DMD patients and are associated with the activation of pseudo-exons and the consequent introduction of premature termination codons [9].

Dystrophin is a large cytoskeletal protein of 3685 amino acids that plays a pivotal role in maintaining the structural integrity of muscle fibers. It is located on the inner face of muscle cell membranes, where it stabilizes the sarcolemma during contraction and relaxation. Dystrophin, in concert with the dystrophin-associated glycoprotein complex (DGC), anchors the actin cytoskeleton to the extracellular matrix (ECM), thereby providing mechanical stability during muscle contraction. The DGC includes key proteins such as dystrophin, the dystroglycan subcomplex (α- and β-dystroglycan), the sarcoglycan subcomplex (α-, β-, γ-, and δ-sarcoglycan), sarcospan, syntrophin, dystrobrevin, and neuronal nitric oxide synthase [3]. Additionally, the DGC is involved in regulating muscle-related gene expression. In the absence of dystrophin, the sarcolemma becomes increasingly fragile, leading to repeated cycles of damage and repair, ultimately resulting in muscle cell necrosis, intrinsic myofibrillar degeneration, oxidative stress, and persistent chronic inflammation within the affected muscle tissue [10]. This pathological condition induces a dysregulation of sarcolemmal ion channels that leads to a significant effect on the excitability and contractility of muscle fibers and cardiomyocytes [11,12]. In this sense, alterations in intracellular calcium management contribute to the severity of the disease in DMD, probably also due to the activation of calcium-activated proteases [8]; furthermore, the high cytoplasmic sodium concentration represents the main cause of the marked edema that is already detectable in boys before fatty degeneration occurs [13].

In cardiac muscle, contraction and relaxation are tightly regulated by calcium cycling between the sarcoplasmic reticulum (SR) and the cytoplasm. In the context of muscular dystrophies, disruptions in calcium homeostasis lead to intracellular calcium overload, which triggers mitochondrial dysfunction including calcium overload and a failure to generate sufficient adenosine triphosphate (ATP). This results in the compromise of calcium pumps, hypercontraction of muscle fibers, and eventual myocyte necrosis [8]. Alterations affecting mitochondria are mainly represented by the reduction in biogenesis and dynamics, decreased intensity of OXPHOS, overgeneration of ROS, and rearrangements of ion transport systems such as a reduction in mitochondrial potassium transport [14]. The pathophysiology of cardiomyopathy in DMD is driven by multiple interrelated mechanisms including sarcolemma instability, calcium dysregulation, altered mitochondrial potassium transport, increased reactive oxygen species (ROS), nitric oxide (NO) dysregulation, and the development of myocardial fibrosis. The cardinal features of cardiomyopathy in DMD include contractile dysfunction, cardiomyocyte death, and the progressive accumulation of fibrotic tissue in the myocardium. A comprehensive understanding of the multiorgan pathophysiology underlying DMD-associated cardiomyopathy is crucial for improving the prognosis, management, and treatment of cardiac complications [15].

DMD predominantly affects males, while female carriers, possessing a single X-linked variant, are generally asymptomatic due to the presence of a second, functional dystrophin allele. However, a subset of female carriers, known as manifest carriers, may exhibit muscle weakness, fatigue, and cardiac involvement. Notably, cardiac symptoms in female DMD carriers affect approximately 8% of this population, with DCM being a common manifestation [16]. The large size of the DMD gene and its susceptibility to a broad spectrum of genetic alterations complicate diagnostic efforts. While multiplex ligation-dependent probe amplification (MLPA) can identify deletions and duplications in over 70% of DMD/BMD cases, the remaining 30% of cases, often caused by single nucleotide variants (SNVs), require additional analysis such as Sanger sequencing [17]. Recently, traditional techniques have been supplemented by high-throughput sequencing technologies such as next-generation sequencing (NGS) including long read sequencing (LRS), which enables molecular diagnosis in 92% of DMD/BMD patients with a single diagnostic approach, making it increasingly suitable for routine clinical practice [17,18].

In parallel, the advent of high-throughput sequencing has led to the identification of modulatory genes that may influence the severity and phenotypic expression of DMD, highlighting the role of the genetic background in predicting disease progression and risk stratification.

## 3. New Frontiers in the Molecular Diagnosis of DMD

The most common cardiac involvement associated with DMD patients is DCM. However, some cases of DMD patients with hypertrophic cardiomyopathy [19,20,21] and left ventricular noncompaction cardiomyopathy [22,23] have been reported. The 2023 European Society of Cardiology (ESC) guidelines [24] recognize more than 20 genes implicated in monogenic, non-syndromic DCM, although robust experimental evidence for most remains limited. Notably, pathogenic variants in lamin A/C and cardiac sarcomere genes are identified in 20–25% of DCM cases. Furthermore, oligogenic inheritance patterns involving multiple potentially causative variants for DCM are increasingly reported [25]. The ESC guidelines also recommend obtaining a three-generation family history for patients with primary cardiomyopathies, conducting clinical screening for first-degree relatives, and offering genetic counseling for affected individuals and their relatives. For pathogenic or likely pathogenic variants, cascade screening for at-risk family members should be considered [26].

The etiology of DCM is highly heterogeneous, with monogenic variants contributing to 30–40% of cases, while polygenic and common variants play a substantial role. Truncating variants in *TTN* are strongly associated with DCM, while mutations in *DSP*, *LMNA*, *MYH7*, *RBM20*, *TNNT2*, *BAG3*, and *FLNC* (the latter being linked to myofibrillar myopathy) are also commonly implicated. Less frequent but notable associations include *DES* (linked to desminopathy), *PLN*, *SCN5A*, and *TNNC1*, the latter contributing to the DCM phenotype in a smaller subset of patients [24]. In light of the diverse genetic landscape underlying DCM, identifying these variants offers promising potential for genetic repair strategies.

A recent study using cardiac MRI to assess late gadolinium enhancement (LGE) patterns in 600 DCM patients has advanced our understanding of genotype-phenotype correlations in DCM, particularly for genes like *TTN*, *DSP*, and *LMNA*. These findings classify DCM-related genetic variants into three categories based on LGE patterns: subepicardial (seen in *DMD*, *DSP*, and *FLNC*), nonspecific (observed in *TTN*, *BAG3*, *LMNA*, and *MYBPC3*), and absent/rare (noted in *TNNT2*, *RBM20*, and *MYH7*). Furthermore, LGE patterns correlate with the risk of major ventricular arrhythmias (MVAs), with certain genotypes associated with higher arrhythmic risk [27].

## 4. DMD Mutations

*DMD* mutations have been extensively studied as a cause of DMD, complicated by cardiac dysfunction. It is unclear why some *DMD* variants result in primary DCM with absent or subclinical skeletal muscle involvement. DMD-associated DCM without severe skeletal myopathy is characterized by incomplete penetrance but a high risk of major adverse cardiovascular events (MACEs) including progression to end-stage HF and ventricular arrhythmias [28].

In the genetic testing of forty males with a potential X-linked genetic cause of primary DCM, four pathogenic/probably pathogenic *DMD* variants were found in five patients. All pathogenic/probably pathogenic *DMD* variants involved dystrophin exons, indicating high levels of constitutive expression in the heart, while fifteen variant-negative DMD probands had variants in autosomal genes including *TTN*, *BAG3*, *LMNA*, and *RBM20*. Distinguishing X-linked causes of DCM from autosomal genes that show sex differences in clinical presentation is critical for informed family management [29]. Truncating variants of *TTN* have been observed in 20% of the studied population as the most common cause of DCM, with extensive alternative splicing. Dystrophin and titin have key roles in cardiac structure and function, and mutations in *DMD* and *TTN* share several pathophysiological mechanisms including changes in force transmission, resistance to mechanical stress, cell signaling, myocardial energetics, and cell survival [29].

Currently, 145 different pathogenic and likely pathogenic variants reported by multiple submitters in patients with DMD have been listed in the ClinVar Registry (July 2024) (Figure 1). As shown in Figure 1, variants are distributed across the entire gene. According to different studies reported in the literature, variant localization may reflect some specific cardiovascular characteristics such as the onset of DCM related to pathogenic variants in *DMD* exons 12 and 14–17 or the presence of heart diseases in patients carrying variants in exons 31–42 [30]. Furthermore, a decrease in cardiac risk has been reported to be associated with variants located in exons 51 or 52 [30]. These data indicate that specific *DMD* variants can lead to cardioprotective or DCM-causing effects in DMD/BMD patients according to the specific location of the genetic alteration [30].

Individuals with the same *DMD* mutation may exhibit differences in disease severity, with modulation by modifier gene polymorphisms in genes distant from the *DMD* gene [31]. Isoleucine-alanine-alanine-methionine (IAAM) homozygosity has been associated with early loss of ambulation resulting in the anti-fibrotic action of the IAAM haplotype of LTBP4 with an apparent protective function from the onset and progression of DMD-related cardiomyopathy [31]. Large genomic mapping studies offer the potential to discover more disease-modifying genes, allowing for the identification of multi-locus interaction patterns and improving the prognosis of the DMD population [31].

Genotype-phenotype correlation studies in BMD patients have revealed that the ABD or R16/R17 nNOS binding domain is often associated with more severe disease compared to BMD patients with other variants; furthermore, in other studies of this type, the R16–R19 regions have been found to be important for cardiac disease. Finally, BMD patients with the same mutation have shown variations in disease severity, highlighting the importance of genetic modifiers, whereby variations in the involved genes can influence disease outcome. Some genetic modifiers that slow disease progression are represented by variants that affect the expression and function of SPP1, LTBP4, CD40, THBS1, ACTN3, and TCTEX1D [3].

## 5. Discussion

Genetic testing has become an invaluable tool in the management of Duchenne cardiomyopathy, evolved significantly in recent years. While it was once primarily used for diagnosis, advances in NGS, Whole exome sequencing (WES), and whole genome sequencing (WGS) now enable genetic tests to provide crucial insights into predicted cardiac involvement and disease progression. These findings are essential for accurate risk stratification, allowing for early identification of individuals at higher risk. This, in turn, facilitates the initiation of protective treatments and the personalization of clinical monitoring, ultimately improving patient outcomes and care.

The identification, management of symptoms, and prevention of disease-related complications (including sudden cardiac death and HF) are the cornerstone of the management of all cardiomyopathies. Overall, genetic testing should be performed in affected individuals, and its results can influence risk stratification and management [24].

Depending on the portion of the genome to analyze, several approaches can be performed. Targeted gene sequencing is a valuable approach to adopt in case of diseases with known disease-causing genes to analyze. Focused panels contain a set of genes that are associated with the inherited disease. Clinical exome sequencing (CES) tests or Mendeliome are able to analyze more than 5000 clinical genes specifically related to known inherited diseases. WES allows for the analysis of all coding regions (2%) of the DNA, and WGS tests all the DNA including intron and exon regions. The latter is the most complex but the most useful for identifying alterations responsible for a disease for which the responsible genes are unknown. However, targeting panels and WGS are the most widely used to evaluate patients affected by DCM. Recently, a linked-read sequencing technology was applied that combined single-molecule barcoding with short-read WGS. A deletion of exons 16–29 in the *DMD* gene was responsible for the disease in the family of a female carrier of X-linked MD with an unsolved genetic status, but she showed a normal dosage of these exons by MLPA and the comparative genomic hybridization (CGH) array, thus usually considering her a “non-carrier”. Unexpectedly, the girl also showed an increased dosage of flanking exons 1–15 and 30–34. Using linked-read WGS, the authors distinguished between the two X chromosomes. In the first allele, they found the 16–29 deletion, while the second allele showed a 1–34 duplication. This duplication in trans restored the normal dosage of exons 16–29 seen by quantitative analyses, converting a non-carrier into a double carrier status prediction [32]. This highlights the significant role of the appropriate choice and research of targeted genetic testing.

The interpretation of results and classification of variants, depending on their association with disease, is the most critical aspect of genetic testing. Misclassification of a variant can lead to excessive testing and unnecessary worry, and consequently, psychosocial impacts on patients and their families. Likewise, misclassification of a variant as benign can lead to false reassurance and the missed diagnosis of a serious disease. Therefore, this risk should be minimized by involving a multidisciplinary team in patient care [33]. The ethical and social implications linked to genetic testing must be taken into consideration. Although the benefits can be of great assistance in DMD patient management, they can also contribute to an increased risk of depression, psychological risks, and the inappropriate use of risk-reducing strategies, especially in disorders related to sudden cardiac death such as the implantation preventive implantable cardioverter-defibrillators (ICDs).

Furthermore, the careful assessment of fragility and comorbidities, with a multidisciplinary approach in DMD patients, is essential. The interaction and collaboration of patients with different healthcare specialists leads to a shared decision-making process, which is necessary for a successful strategy. Knowledge of the patient-specific genetic background, using genetic tests, is a potentially useful tool in the diagnosis and prognosis of DMD-associated cardiomyopathy in the direction of predictive, precision, and personalized medicine. Due to the importance of this powerful approach, in the last years, a newborn screening (NBS) program for DMD has been developed in many countries. Furthermore, to facilitate early diagnosis and intervention before the onset of observable clinical disease, a biochemical test for creatine kinase (CK) activity, which is elevated in individuals with DMD, is performed before genetic testing [34].

## 6. Future Outlooks

Artificial intelligence (AI) subsets of machine learning (ML) and deep learning could provide more accurate stratification and typing of DMD patients, offering a potential solution to optimize personalized medicine. The development of effective knowledge systems that integrate different levels of data is important to maximize the impact on translational research [35]. The emerging impact of the application of AI can form the basis for the storage and management of genetic data that can be correlated with the clinical and biological characteristics of the patient’s treatment path. For example, the application of AI to wearable devices on DMD patients has allowed the use of ML algorithms to accurately predict the progression of the disease and their personal trajectories [36]. The application of this methodology, which considers the patient’s specific genetic background, could lead to a complex and personalized management model of patients with DMD-associated cardiomyopathy and have a notable impact on both patient care and the burden it places on the national health system. These results will likely be obtained in the future through genomic characterization of the patient by integrating genomic data with other clinical data such as cardiac imaging, coronary angiography, and clinical biomarkers [37]. Furthermore, advances in big data analytics approaches and etiopathogenesis studies could help identify the key origin of the disease’s severe phenotypic variability in both the onset and progression of DMD-associated cardiomyopathy.

In conclusion, the integration of advanced genetic strategies with cutting-edge AI tools offers unprecedented opportunities to transform the way we understand and manage cardiac involvement in Duchenne Muscular Dystrophy. These technologies are not only paving the way for more precise risk stratification but also for personalized treatment plans that could significantly improve patient outcomes. As we venture into this new era, it is clear that the potential for innovation is limitless.

As Albert Einstein once said, “The important thing is not to stop questioning. Curiosity has its own reason for existing”. In this spirit, the relentless pursuit of knowledge and technological advancement will continue to shape the future of DMD research and treatment.

## Figures and Tables

**Figure 1 biomedicines-13-00102-f001:**
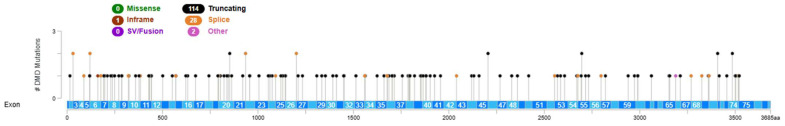
Distribution of pathogenic and likely pathogenic variants identified in patients referred for DMD genetic testing submitted in ClinVar.

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
