# Peer review of "Unraveling the Genetic Heartbeat: Decoding Cardiac Involvement in Duchenne Muscular Dystrophy"

_biomedicines, 2025, doi:10.3390/biomedicines13010102_

Round 1

Reviewer 1 Report

Comments and Suggestions for Authors

This relatively short and well-focused review summarizes the latest data on the nature of cardiomyopathy in Duchenne muscular dystrophy (DMD). This is an important topic because skeletal muscle pathology is now amenable to some treatment options in DMD, while cardiac pathology remains an important cause of patient mortality. I have the following comments about the work:

1. Lines 130-139. It should be noted that dystrophin and DAPC are also involved in the regulation of sarcolemmal ion channels and the loss of its components leads to dysregulation of a number of ion channels (calcium, potassium, sodium, etc.) and this has a significant effect on the excitability and contractility of muscle fibers or cells in the case of the heart muscle. Moreover, excess calcium leads to the activation of proteases and phospholipases, sodium also contributes to edema, this is well demonstrated. These changes are an important intermediate link leading to the phenomena described in lines 136-139, they should not be forgotten.

2. Lines 141-144. It should also be noted that there are changes in the mitochondrial transport of potassium ions, which seems to contribute to the overproduction of ROS in the cell. I also think that in this same paragraph it is possible to note changes in the interaction of the endoplasmic reticulum and mitochondria, which is important for the regulation of ion transport (primarily calcium), ROS generation, etc.

3. A small remark regarding the discussion. Recently, the following approach has been used to manage and treat patients with genetic pathologies, including DMD. After genetic analysis of the patient, a laboratory model (cells, animals) with an identical mutation is created. These models are used to evaluate the effectiveness and safety of genetic approaches (vectors, etc.) for further use in the clinic. What do the authors think about this approach?

Author Response

Thank you very much for taking the time to review this manuscript. Please find the detailed responses below and the corresponding revisions in track changes in the re-submitted files.

Point-by-point response to Comments and Suggestions

Comments 1: Lines 130-139. It should be noted that dystrophin and DAPC are also involved in the regulation of sarcolemmal ion channels and the loss of its components leads to dysregulation of a number of ion channels (calcium, potassium, sodium, etc.) and this has a significant effect on the excitability and contractility of muscle fibers or cells in the case of the heart muscle. Moreover, excess calcium leads to the activation of proteases and phospholipases, sodium also contributes to edema, this is well demonstrated. These changes are an important intermediate link leading to the phenomena described in lines 136-139, they should not be forgotten.

Response 1: As suggested, we mentioned in the text (Pag. 3; lines 141-147) the alterations of DMD ion channels related as an intermediate link leading to the phenomena described in the following lines.

Comments 2: Lines 141-144. It should also be noted that there are changes in the mitochondrial transport of potassium ions, which seems to contribute to the overproduction of ROS in the cell. I also think that in this same paragraph it is possible to note changes in the interaction of the endoplasmic reticulum and mitochondria, which is important for the regulation of ion transport (primarily calcium), ROS generation, etc.

Response 2: We have added in the text (Page.4; lines 153-156) the alterations affecting the mitochondria such as the altered mitochondrial transport of potassium.

Comments 3: A small remark regarding the discussion. Recently, the following approach has been used to manage and treat patients with genetic pathologies, including DMD. After genetic analysis of the patient, a laboratory model (cells, animals) with an identical mutation is created. These models are used to evaluate the effectiveness and safety of genetic approaches (vectors, etc.) for further use in the clinic. What do the authors think about this approach?

Response 3: Actually, hIPSc is the most valuable model to understand the role of the genetic variants identified in patients. This is an important step for the development of new therapeutic personalize strategies in DMD patients.

Reviewer 2 Report

Comments and Suggestions for Authors

The authors reviewed molecular genetics and pathogenesis of Duchenne Muscular Dystrophy (DMD). The paper’s structure and content can be improved.

From the title, genetics and pathogenesis are assumed to be the theme of this review. However, the introduction is heavy on clinical manifestations and treatment protocols. It would be good to set the tone so readers can expect what the paper is about. I also expected the paper to be mainly about DMD cardiomyopathy, but the main section is just DMD. Please ensure the title matches with the theme.

Section 3 is “New Frontiers in the Molecular Diagnosis of DMD,” but the content is all about DCM. As a review of DMD cardiomyopathy, common types of cardiomyopathies in DMD patients should be addressed.

Newborn DMD screening has been available in many regions of the world. Creatine kinase blood test is often performed before genetic testing. This should be addressed.

Section 4, “DMD Mutations,” stated that there were 145 pathogenic/likely pathogenic variants by multiple submitters in ClinVar, but I see more than 400 records in ClinVar. Did you use other criteria?

Author Response

Thank you very much for taking the time to review this manuscript. Please find the detailed responses below and the corresponding revisions in track changes in the re-submitted files.

Point-by-point response to Comments and Suggestions

Comments 1: From the title, genetics and pathogenesis are assumed to be the theme of this review. However, the introduction is heavy on clinical manifestations and treatment protocols. It would be good to set the tone so readers can expect what the paper is about. I also expected the paper to be mainly about DMD cardiomyopathy, but the main section is just DMD. Please ensure the title matches with the theme.

Response 1: Thank you for the comments. We agree with you and change the title to: “Architecture of DMD: a spotlight on cardiomyopathies.”

Comments 2: Section 3 is “New Frontiers in the Molecular Diagnosis of DMD,” but the content is all about DCM. As a review of DMD cardiomyopathy, common types of cardiomyopathies in DMD patients should be addressed.

Response 2: Cardiomyopathy in DMD patients is present as dilated cardiomyopathy that progressively increases with age with diffuse hypokinesia. In particular, DMD muscle is characterized by myonecrosis, reactive myelofibrosis, fatty substitution and chronic inflammation. Only few cases of DMD with Hypertrophic Cardiomyopathy are reported in the literature. Furthermore, in the 2023 ESC Guidelines, dystrophinopathies are reported exclusively in relation to the dilated cardiomyopathy (DCM) phenotype. As reported in the ESC Guidelines, among the examples of inheritance patterns, signs and symptoms, and electrocardiographic features that should raise suspicion of specific etiologies, grouped according to the cardiomyopathy phenotype, such as HCM, DCM, NDLVC, AVC-AR, and RCM, dystrophinopathies are reported in the DCM group (https://doi.org/10.1093/eurheartj/ehad194). We have added in Paragraph 3 (Pag.4; lines 186-188) as follows:

“The most common cardiac involvement associated with DMD patients is DCM. However, some cases of DMD patients with Hypertrophic Cardiomyopathy (Greiner, et al.; Aspit, L., et al; Tandon, A, et al) and Left ventricular noncompaction cardiomyopathy (Parent, J. J., et al; Statile, C. J., et al) have been reported.”

Comments 3: Newborn DMD screening has been available in many regions of the world. Creatine kinase blood test is often performed before genetic testing. This should be addressed.

Response 3: As suggested, we have added a reference to neonatal screening for DMD in the Discussion (Pag. 7; lines 308-312).

Comments 4: Section 4, “DMD Mutations,” stated that there were 145 pathogenic/likely pathogenic variants by multiple submitters in ClinVar, but I see more than 400 records in ClinVar. Did you use other criteria?

Response 4: 145 variants reported in ClinVar have been filtered according to these criteria:

- Pathogenic and Likely Pathogenic;

- Associated only to Condition Duchenne Muscular Dystrophy;

- Reported by multiple submitters.

Round 2

Reviewer 1 Report

Comments and Suggestions for Authors

The authors responded adequately to my comments. One remark: the authors made a mistake in the order of references 13-16, this needs to be brought into line with the text.

Author Response

The authors responded adequately to my comments. One remark: the authors made a mistake in the order of references 13-16, this needs to be brought into line with the text.

We thank the Reviewer for his kind observation. We have deleted Reference 15 from the previous Version, as it was reported incorrectly.

Reviewer 2 Report

Comments and Suggestions for Authors

The authors have addressed most of my questions.

Please do not use abbreviations in the title.

I used the same criteria described by the authors and still found about 400 records in ClinVar. You may want to include the date of your data acquisition.

Author Response

The authors have addressed most of my questions.

Please do not use abbreviations in the title.

Thank you for the comment. We have removed the abbreviation from the Title.

I used the same criteria described by the authors and still found about 400 records in ClinVar. You may want to include the date of your data acquisition.

The number of the variants reported follows this path: apply the filter Pathogenic and Likely Pathogenic and Reported by multiple submitters, you will have 480 variants. Then we downloaded the file and from there we filtered for Associated only to Condition Duchenne Muscular Dystrophy. We also included the date of the search in ClinVar (Page 5, Line 235).
